# Simulation and Exergy Analysis of a Refrigeration System Using an Open-Source Web-Based Interactive Tool—Comparison of the Conventional Approach and a Novel One for Avoidable Exergy Destruction Estimation

**Volodymyr Voloshchuk** [1,*] **, Paride Gullo** [2] **, Eugene Nikiforovich** [3] **and Nadia Buyak** [4]

1 Department of Automation of Thermal Processes, National Technical University of Ukraine "Igor Sikorsky Kyiv Polytechnic Institute", Prosp. Peremohy 37, 03056 Kyiv, Ukraine
2 Department of Mechanical and Electrical Engineering, University of Southern Denmark (SDU), Alsion 2, 6400 Sønderborg, Denmark; parigul@sdu.dk
3 Department of Nuclear Power Plants and Engineering Thermal Physics, National Technical University of Ukraine "Igor Sikorsky Kyiv Polytechnic Institute", Prosp. Peremohy 37, 03056 Kyiv, Ukraine; eugenen@kth.se
4 Department of Thermal Engineering and Energy Saving, National Technical University of Ukraine "Igor Sikorsky Kyiv Polytechnic Institute", Prosp. Peremohy 37, 03056 Kyiv, Ukraine; n.buyak@kpi.ua
* Correspondence: v.voloshchuk@kpi.ua

**Abstract:** Avoidable endogenous/exogenous parts of the exergy destruction in the components of an energy conversion system can be computed by applying advanced exergy analysis. Their calculation is crucial for the correct assessment of the real thermodynamic enhancement achievable by the investigated energy conversion system. This work proposes a new approach to estimate the avoidable exergy destruction rates of system components, being more rigorous compared to the conventional method due to the elimination of the need for the implementation of theoretical assumptions associated with the idealization of processes. An open-source web-based interactive tool was implemented to contrast the results of the conventional advanced exergy analysis to those involving the new approach for avoidable exergy destruction estimation. The comparison was based on the same case study, i.e., a refrigeration system selected from the literature. It was observed that the developed tool can be properly employed for comparing the two approaches within exergy analyses, and the results obtained presented some differences for the compressor and the condenser. Compared to the new approach, the existing methodology of advanced exergy analysis suggests lower values of the avoidable part of exergy destruction, which can be reduced by improving the efficiency of the compressor and the condenser. Moreover, the avoidable parts of exergy destruction, which could be removed within these components by improving the efficiencies of the remaining components, were higher in the case of the application of the existing advanced exergetic analysis as compared with the findings obtained by the proposed approach. These differences were due to the impossibility of the existing advanced exergy analysis to implement complete thermodynamic "idealization" for the condenser and evaporator.

**Keywords:** advanced exergetic analysis; avoidable exergy destruction; open-source tool; refrigeration system

## 1. Introduction

The distinguishing feature of exergy analysis is that, unlike energy assessment, it is a more powerful tool for the investigation and the performance improvement of energy conversion systems. In addition to conventional exergy evaluation, the so-called advanced exergy analysis has been developed at the Institute for Energy Engineering of Technische Universität Berlin in the last twenty years [1–3]. The advanced exergy method can identify

the interactions among the components of the investigated system and reveal the real potential for the enhancement of the individual components, as well as of the overall energy conversion system. In the last several years, advanced exergy analysis has been demonstrated in several applications. Ozcan et al. [4] investigated a district heating system based on four types of waste heat sources: municipal solid waste cogeneration, thermal power, wastewater treatment, and cement production. The results obtained showed that the exergy destruction of the pump was mainly avoidable (endogenous), and it was mainly unavoidable for the heat exchangers. Gullo [5] applied advanced exergy analysis for estimation of the thermodynamic performance of a transcritical R744 booster supermarket refrigeration system equipped with a R290 dedicated mechanical subcooling system. The author observed that the approach temperature of the gas cooler as well as the outdoor temperature influenced its avoidable irreversibilities significantly. Voloshchuk et al. [6] found that 63% and 20% of the avoidable exergy destruction occurring in a heat pump system for space heating purposes was caused by the evaporator and the condenser, respectively. Dibazar et al. [7] studied the performance of three typologies of organic Rankine cycles (i.e., basic cycle, cycle with single regeneration, and cycle with double regeneration) for the condition of an unchanged heat source. The outcomes obtained revealed that the regenerative cycles have high potential to decrease the irreversibilities compared to the basic cycle. Sun and Liu [8] proposed a novel transcritical $CO_2$ energy storage-based trigeneration system. The implementation of advanced exergy analysis brought to light the need for the enhancement of the cold storage and the two compressors. Tinoco-Caicedo et al. [9] found that the spray dryer needs to be enhanced in order to improve the performance of industrial-scale spray drying processes for the production of instant coffee. Mortazavi and Ameri [10] demonstrated that solar flat plate air collectors can be enhanced by improving their glass cover. Ehyaei et al. [11] highlighted the increase in avoidable exergy destruction for higher wind speeds.

According to advanced exergy analysis, the exergy destruction of each component belonging to the investigated energy conversion system can be split into several parts. The so-called unavoidable, endogenous, and unavoidable endogenous parts of the exergy destruction can be calculated assuming special operating conditions of the components. The remaining ones (i.e., avoidable, exogenous, unavoidable exogenous, avoidable exogenous, and avoidable endogenous parts) are then calculated as differences between the appropriate parts of the exergy destruction. Avoidable exogenous and avoidable endogenous exergy destruction, being the most important parts from a practical viewpoint, cannot be obtained without estimating the endogenous and unavoidable endogenous parts of the exergy destruction first.

The avoidable exergy destruction evaluates the real potential for improving a system component. The unavoidable one is the part of the exergy destruction that cannot be further reduced due to technological limitations, such as the availability and/or costs of materials and manufacturing methods. For calculating the unavoidable exergy destruction rates within each system component, the "best" operation conditions, which cannot be realized in the foreseeable future, are assumed. The calculation of the avoidable exergy destruction is associated with arbitrary decisions. Within advanced exergy analysis, the unavoidable exergy destruction can be estimated following two approaches: at the component level [12,13] and at the overall system level [2,12,14]. If the overall system-level approach is applied, the minimum thermodynamic inefficiencies ("best" operation conditions) are introduced simultaneously for all the components of the system. Such a method allows the engineers to compute the unavoidable exergy values of each component by simulating the overall system only one time. The overall system-level approach should be applied only for quite simple systems, such as refrigeration/heat pump units, simple open cycle gas turbine systems, etc. It is impossible to use this method for complex systems [3,12]. The component-level approach means that, in order to calculate the values of unavoidable exergy destruction, each component needs to be simulated individually while it is operat-

ing at its "best" operation conditions. For the implementation of this approach, relatively simple calculations can be applied separately for each component [3,12].

The endogenous exergy destruction is a part of the exergy destruction taking place within the k-th component as the considered component is operating with its actual efficiency and all other components are working ideally (i.e., their exergy efficiency is equal to 1). Some approaches for splitting the exergy destruction into endogenous and exogenous parts have been developed, such as the thermodynamic cycle method [12,15,16], the engineering approach [15–17], the exergy balance method [15], the equivalent component method [15], the method of structural coefficients [15], and malfunction/dysfunction analysis [18–20]. A detailed description and comparison of these methods can be found in [15]. Additionally, the authors in [21] developed a new approach (i.e., decomposition method) for the estimation of the endogenous parts of exergy destruction, which is easier and faster compared to the previous approaches [21,22]. The unavoidable endogenous exergy destruction is estimated similarly, but under the assumption that the studied component is operating with its unavoidable thermodynamic inefficiencies.

Estimation of the endogenous exergy destruction requires the implementation of theoretical assumptions associated with the idealization of processes, which is one of the most critical issues. For example, the throttling process is always irreversible and it is not possible to present it as an ideal process. As a result, the throttling valve is replaced by an ideal expansion device [2,14]. As for the heat exchangers, the complete thermodynamic "idealization" is not possible either. Only the conditions for which the pinch point temperature is equal to zero (but exergy destruction is not equal zero) can be suggested [2,14]. As a result, in the case of the exergy analysis of a refrigeration system, it is impossible to completely calculate the endogenous part of exergy destruction within the compressor because it is impossible to consider ideal processes (i.e., without any thermodynamic inefficiencies) within the evaporator and the condenser. The same holds true for the calculation of the "pure" endogenous part of the exergy destruction within the condenser due to the inclusion of some parts of irreversibilities within the evaporator. Similar drawbacks can be experienced as the exergy destruction of both the evaporator and the throttling valve is assessed.

The scope of the paper is to develop and apply an approach to calculating the avoidable parts of exergy destruction within the k-th component without the need for the implementation of the aforementioned theoretical assumptions, which are associated with the idealization of processes or plant components. An open-source, web-based, interactive computing tool for applying both the conventional advanced exergy analysis and the new approach for the estimation of the avoidable exergy destruction to a refrigeration system is developed. The two considered advanced exergy methodologies are described in Section 2, while the results obtained are presented, discussed, and compared in Section 3. Finally, the conclusions are highlighted in Section 4.

## 2. Materials and Methods

The existing methodology of the advanced exergy analysis splits exergy destruction within each system component into the following parts [2,12,14]:

- endogenous/exogenous parts

$$\dot{E}_{D,k} = \dot{E}_{D,k}^{EN} + \dot{E}_{D,k}^{EX} \tag{1}$$

- unavoidable/avoidable parts

$$\dot{E}_{D,k} = \dot{E}_{D,k}^{AV} + \dot{E}_{D,k}^{UN} \tag{2}$$

- and combined parts

$$\dot{E}_{D,k} = \dot{E}_{D,k}^{UN,EN} + \dot{E}_{D,k}^{UN,EX} + \dot{E}_{D,k}^{AV,EN} + \dot{E}_{D,k}^{AV,EX} \tag{3}$$

The thermodynamic cycle-based approach was used to split the exergy destruction into parts [12,15,16]. The product exergy rate (i.e., the cooling capacity) of the investigated refrigeration system in all the analyzed cycles remained the same.

The endogenous exergy destruction ($\dot{E}_{D,k}^{EN}$, being associated with the k-th component, is caused only by the actual irreversibilities occurring in the same component as all other components operate in an ideal way. For estimating the endogenous part of the exergy destruction within each component of the refrigeration system, the hybrid cycles with only one irreversible component were analyzed. The separate introduction of irreversibilities within each system component provides the possibility to calculate the endogenous exergy destruction for each component. The exogenous part of exergy destruction ($\dot{E}_{D,k}^{EX}$) within the k-th component is related to the irreversibilities occurring in the remaining components or caused by the structure of the overall system. $\dot{E}_{D,k}^{EX}$ in the k-th component was estimated by subtracting the endogenous exergy destruction from the total one [2,12,14].

For determining the unavoidable and avoidable parts of exergy destruction, hybrid cycles with unavoidable thermodynamic inefficiencies occurring within each component needed to be implemented. The avoidable exergy destruction ($\dot{E}_{D,k}^{AV}$) was calculated as the difference between the total and unavoidable parts of exergy destruction [2,12,14].

For further splitting exergy destruction, the thermodynamic cycle-based approach was applied only for the estimation of the unavoidable endogenous exergy destruction ($\dot{E}_{D,k}^{UN,EN}$) [2,12]. A similar approach applied for calculating the endogenous part of the exergy destruction could be used for this purpose. However, for this case, the unavoidable thermodynamic efficiency of each component was assumed [2,12,14]. The remaining parts of the exergy destruction were then calculated as [2]

$$\dot{E}_{D,k}^{UN,EX} = \dot{E}_{D,k}^{UN} - \dot{E}_{D,k}^{UN,EN} \tag{4}$$

$$\dot{E}_{D,k}^{AV,EN} = \dot{E}_{D,k}^{EN} - \dot{E}_{D,k}^{UN,EN} \tag{5}$$

$$\dot{E}_{D,k}^{AV,EX} = \dot{E}_{D,k}^{AV} - \dot{E}_{D,k}^{AV,EN} \tag{6}$$

According to [2,3], the endogenous avoidable part of exergy destruction ($\dot{E}_{D,k}^{AV,EN}$) can be decreased by improving the efficiency of the considered component. The exogenous avoidable part of the exergy destruction ($\dot{E}_{D,k}^{AV,EX}$) can be reduced by a structural improvement of the overall system or by increasing the efficiencies of the remaining components.

Taking into account that the investigators need to consider possibilities for the reduction of the avoidable parts of exergy destruction occurring in each system component, this work was based uniquely on these irreversibilities and proposes a new calculation method to compute them. According to the novel method, the avoidable exergy destruction rate internally caused ($\dot{E}_{D,k}^{AV,INT}$) could be computed as the difference between the total exergy destruction of the investigated component ($\dot{E}_{D,k}$), i.e., calculated under real operation conditions, and its exergy destruction ($\dot{E}_{D,k}^{MIN,k}$) evaluated under conditions at which its irreversibilities were reduced by improving its efficiency, taking into account that the remaining components were operating under real conditions

$$\dot{E}_{D,k}^{AV,INT} = \dot{E}_{D,k} - \dot{E}_{D,k}^{MIN,k} \tag{7}$$

The avoidable exergy destruction within the k-th component, being caused by the avoidable irreversibilities occurring within the rest of the components (i.e., externally caused) $\dot{E}_{D,k}^{AV,EXT}$, could be computed by subtracting the exergy destruction rate ($\dot{E}_{D,k}^{MIN,rest}$) within the k-th component under conditions at which the remaining components were

working with reduced irreversibilities from the exergy destruction rate ($\dot{E}_{D,k}$) taking place within the k-th component under real operation

$$\dot{E}_{D,k}^{AV,EXT} = \dot{E}_{D,k} - \dot{E}_{D,k}^{MIN,rest} \tag{8}$$

It could be accepted that the avoidable endogenous and avoidable exogenous parts of exergy destruction (Equations (5) and (6)) allow the calculation of the same exergy destruction parts as the ones determined by the authors as avoidable internally and externally caused (Equations (7) and (8)).

The schematic of the investigated vapor-compression refrigeration system, consisting of the compressor (CM), the condenser (CD), the throttling valve (TV), and the evaporator (EV), is illustrated in Figure 1.

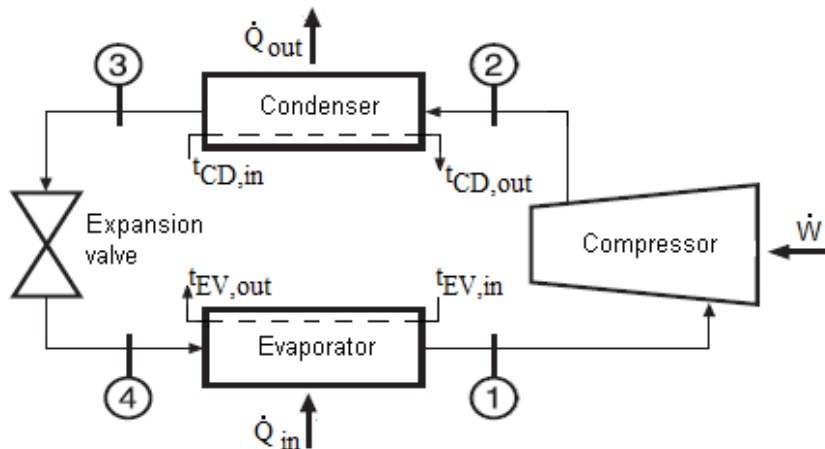

**Figure 1.** Schematic of the investigated vapor-compression refrigeration system: 1–4—stream numbers.

The analysis was performed for a refrigeration system having a cooling capacity of 100 kW [2]. Air was used as a secondary working fluid for the both the evaporator and condenser. The following assumptions were made [2]: (i) in the evaporator, air was cooled from −5 °C down to −15 °C; (ii) in the condenser, ambient air was heated from 20 °C up to 30 °C; (iii) the minimal temperature difference within the evaporator (i.e., between air and refrigerant) and within the condenser (i.e., between refrigerant and air) was 10 K. The real isentropic efficiency of the compressor was assumed to be 0.80 [2]. For evaluating the unavoidable exergy destruction rates (for applying the existing advanced exergy analysis methodology), the following assumptions were made: the unavoidable temperature differences in the evaporator and the condenser were equal to 0.50 K, whereas the unavoidable compressor efficiency was equal to 0.95. For a fair comparison of the two considered approaches, the same values of the component parameters were assumed for the calculation of internally and externally caused avoidable exergy destruction rates. For creating the theoretical cycle of the studied refrigeration system within the existing advanced exergy analysis methodology, the following assumptions were used [2]: (i) for both the evaporator and condenser, the minimal temperature differences were equal to 0 K; (ii) the compressor efficiency was equal to 1.0; (iii) an ideal expansion process was used for replacing the throttling valve. An ambient temperature equal to 20 °C was chosen as the reference point for the exergy analysis [23].

An open-source, web-based, interactive computing tool was developed [24] and applied for the advanced exergy analysis of the investigated refrigeration system. The tool was utilized by employing (i) a cloud service for developing notebooks using the object-oriented programming language Python and (ii) CoolProp [25], being a library for calculating the thermophysical properties of working fluids.

Results obtained from the authors' model were compared with the ones presented in [2] using the relative difference of estimation

$$\delta = \frac{x - x_{[2]}}{x_{[2]}} \cdot 100\% \tag{9}$$

where $x$ is a value of the parameter obtained on the basis of the model developed by the authors and $x_{[2]}$ is a value of the parameter presented in [2].

## 3. Results and Discussion

The values of specific physical (PH) exergy of streams obtained on the basis of the developed model are compared with the ones presented in [2] in Table 1.

**Table 1.** Values of specific physical exergy of the secondary working fluids (air).

| Stream | $e^{PH}$, kJ·kg$^{-1}$ | | |
|---|---|---|---|
| | Authors' Data | Morosuk and Tsatsaronis [2] | $\delta$, % |
| EV, in | 1.137 | 1.138 | −0.090 |
| EV, out | 2.285 | 2.285 | 0.000 |
| CD, in | 0.000 | 0.000 | 0.000 |
| CD, out | 0.168 | 0.168 | 0.000 |

For an open system, the physical exergy associated with a flowing stream of matter, having a specific enthalpy $h$ and a specific entropy $s$, is calculated as

$$e^{PH} = (h - h_0) - T_0(s - s_0) \tag{10}$$

where $h_0$ and $s_0$ are the specific enthalpy and the specific entropy, respectively, of the same stream calculated at the dead state temperature ($T_0$) and dead state pressure ($p_0$). In this work, $T_0$ and $p_0$ were taken as +20 °C and 0.1 MPa, respectively.

It could be observed that the relative difference of −0.090% was obtained for the physical exergy of the air at the evaporator inlet. Moreover, the other results were equal to the data presented in [2].

The thermodynamic data (i.e., absolute temperature, pressure, and specific physical exergy) of the real cycle for the R134a refrigeration system are presented in Table 2. The values of absolute temperature estimated on the developed model completely coincide with the ones presented in [2]. As for the values of absolute pressure and physical exergy, the maximum relative difference between the authors' model and the one developed in [2] was equal to −0.47 and −0.44%, respectively.

**Table 2.** Thermodynamic values of the real cycle with R134a.

| Stream | $T$, °C | | | $p$, bar | | | $e^{PH}$, kJ·kg$^{-1}$ | | |
|---|---|---|---|---|---|---|---|---|---|
| | Author's Data | Morosuk and Tsatsaronis [2] | $\delta$, % | Authors' Data | Morosuk and Tsatsaronis [2] | $\delta$, % | Authors' Data | Morosuk and Tsatsaronis [2] | $\delta$, % |
| 1 | −25.00 | −25.00 | 0.00 | 1.060 | 1.065 | −0.47 | 4.56 | 4.58 | −0.44 |
| 2 | 61.10 | 61.10 | 0.00 | 10.17 | 10.17 | 0.00 | 52.96 | 52.97 | −0.02 |
| 3 | 40.00 | 40.00 | 0.00 | 10.17 | 10.17 | 0.00 | 40.40 | 40.40 | 0.00 |
| 4 | −25.00 | −25.00 | 0.00 | 1.060 | 1.065 | −0.47 | 27.60 | 27.60 | 0.00 |

The mass flow rates of the air as a secondary working fluid for the real cycle of the R134a refrigeration system calculated on the basis of the developed model contrast those

shown in [2], as demonstrated in Table 3. It was found that the relative differences of estimation do not exceed +0.04%.

**Table 3.** Mass flow rates of working fluids for the real cycle with R134a.

| Working Fluid | $\dot{m}$ kg·s$^{-1}$ | | |
|---|---|---|---|
| | Author's Data | Morosuk and Tsatsaronis [2] | $\delta$, % |
| R134a | 0.787 | 0.787 | 0.00 |
| Air (evaporator) | 9.945 | 9.942 | 0.03 |
| Air (condenser) | 14.546 | 14.540 | 0.04 |

The values of evaporator and condenser temperatures for the theoretical cycle and for the cycle with unavoidable exergy destruction are shown in Tables 4 and 5, respectively. It could be observed that the relative differences between the authors' model and the outcomes from [2] do not exceed +1.35%.

**Table 4.** Temperatures of evaporation and condensation for the theoretical refrigeration cycle.

| Stream | $T$, °C | | |
|---|---|---|---|
| | Author's Data | Morosuk and Tsatsaronis [2] | $\delta$, % |
| Evaporator temeprature | −15.00 | −15.00 | 0.00 |
| Condenser temperature | 30.00 | 29.60 | 1.35 |

**Table 5.** Temperatures of evaporation and condensation for the refrigeration cycle with unavoidable exergy destruction.

| Stream | $T$, °C | | |
|---|---|---|---|
| | Author's Data | Morosuk and Tsatsaronis [2] | $\delta$, % |
| Evaporator temeprature | −15.50 | −15.50 | 0.00 |
| Condenser temperature | 30.50 | 30.10 | 1.33 |

The results above show that the model developed within this work is acceptable for the comparison of the results of the two aforementioned approaches for the estimation of avoidable parts of exergy analyses.

Table 6 presents the results taken from [2] for the advanced exergy analysis of the R134a refrigeration system, while the ones obtained from the authors' model are listed in Table 7.

The comparison of the results from the advanced exergy analyses of the vapor-compression refrigeration machine using R134a is shown in Table 8. It could be observed that most of the values in Table 8 for the relative difference do not exceed ±2.0%. However, the differences for $\dot{E}_{D,CD}^{UN}$ $\dot{E}_{D,CD}^{UN,EN}$, and $\dot{E}_{D,CD}^{UN,EX}$ were above 7.0%. Such a high difference could be explained in the following way. The unavoidable exergy destruction depends on the pinch point temperature ($\Delta T_{CD}$) within the condenser. The assumed value of this temperature difference was equal to 0.5 K. For the sake of simplicity, this value of temperature difference was defined as the difference between the temperature of R134a on the saturated liquid line for conditions of the condenser and the temperature of the heated air at the condenser outlet (see Figure 2). For such assumptions, the condenser temperature was equal to 30.5 °C. In [2], the condensation temperature was 30.1 °C. As a result, the value of unavoidable exergy destruction obtained in this work was higher than the one presented in [2]. A similar explanation could be applied to cases involving $\dot{E}_{D,k}^{UN,EN}$ and $\dot{E}_{D,k}^{UN,EX}$.

**Table 6.** Results obtained from the conventional advanced exergy analysis [2].

| Component | $\dot{E}_{F,k}$, kW | $\dot{E}_{P,k}$, kW | $\dot{E}_{D,k}$, kW | $\dot{E}_{D,k}^{UN}$, kW | $\dot{E}_{D,k}^{AV}$, kW | $\dot{E}_{D,k}^{EN}$, kW | $\dot{E}_{D,k}^{EX}$, kW | $\dot{E}_{D,k}^{UN,EN}$, kW | $\dot{E}_{D,k}^{UN,EX}$, kW | $\dot{E}_{D,k}^{AV,EN}$, kW | $\dot{E}_{D,k}^{AV,EX}$, kW |
|---|---|---|---|---|---|---|---|---|---|---|---|
| CM | 46.360 | 38.090 | 8.271 | 1.097 | 7.174 | 4.850 | 3.421 | 1.031 | 0.066 | 3.819 | 3.355 |
| CD | 9.893 | 2.441 | 7.452 | 2.135 | 5.317 | 5.798 | 1.654 | 2.008 | 0.127 | 3.790 | 1.527 |
| TV | 30.650 | 20.580 | 10.070 | 4.213 | 5.857 | 4.002 | 6.068 | 4.002 | 0.211 | 0.000 | 5.857 |
| EV | 18.130 | 11.410 | 6.714 | 2.358 | 4.356 | 6.714 | 0.000 | 2.358 | 0.000 | 4.356 | 0.000 |

**Table 7.** Results obtained from the advanced exergy analysis based on the authors' model.

| Component | $\dot{E}_{F,k}$, kW | $\dot{E}_{P,k}$, kW | $\dot{E}_{D,k}$, kW | $\dot{E}_{D,k}^{UN}$, kW | $\dot{E}_{D,k}^{AV}$, kW | $\dot{E}_{D,k}^{EN}$, kW | $\dot{E}_{D,k}^{EX}$, kW | $\dot{E}_{D,k}^{UN,EN}$, kW | $\dot{E}_{D,k}^{UN,EX}$, kW | $\dot{E}_{D,k}^{AV,EN}$, kW | $\dot{E}_{D,k}^{AV,EX}$, kW |
|---|---|---|---|---|---|---|---|---|---|---|---|
| CM | 46.368 | 38.096 | 8.272 | 1.109 | 7.163 | 4.897 | 3.375 | 1.041 | 0.068 | 3.856 | 3.307 |
| CD | 9.885 | 2.441 | 7.444 | 2.290 | 5.153 | 5.790 | 1.653 | 2.154 | 0.136 | 3.637 | 1.517 |
| TV | 30.661 | 20.584 | 10.076 | 4.307 | 5.769 | 4.093 | 5.983 | 4.093 | 0.214 | 0.000 | 5.769 |
| EV | 18.134 | 11.414 | 6.720 | 2.365 | 4.356 | 6.720 | 0.000 | 2.365 | 0.000 | 4.356 | 0.000 |

**Table 8.** Comparison of the results obtained from the conventional advanced exergy analysis and those obtained on the basis of the approach proposed in this work.

| Component | Relative Difference, % | | | | | | | | | | |
|---|---|---|---|---|---|---|---|---|---|---|---|
| | $\dot{E}_{F,k}$ | $\dot{E}_{P,k}$ | $\dot{E}_{D,k}$ | $\dot{E}_{D,k}^{UN}$ | $\dot{E}_{D,k}^{AV}$ | $\dot{E}_{D,k}^{EN}$ | $\dot{E}_{D,k}^{EX}$ | $\dot{E}_{D,k}^{UN,EN}$ | $\dot{E}_{D,k}^{UN,EX}$ | $\dot{E}_{D,k}^{AV,EN}$ | $\dot{E}_{D,k}^{AV,EX}$ |
| CM | 0.016 | 0.015 | 0.012 | 1.076 | −0.151 | 0.976 | −1.354 | 0.991 | 2.397 | 0.972 | −1.428 |
| CD | −0.080 | 0.010 | −0.110 | 7.276 | −3.076 | −0.131 | −0.037 | 7.265 | 7.446 | −4.049 | −0.659 |
| TV | 0.035 | 0.021 | 0.063 | 2.233 | −1.498 | 2.272 | −1.393 | 2.272 | 1.506 | - | −1.498 |
| EV | 0.023 | 0.033 | 0.096 | 0.282 | −0.004 | 0.096 | - | 0.282 | - | −0.004 | - |

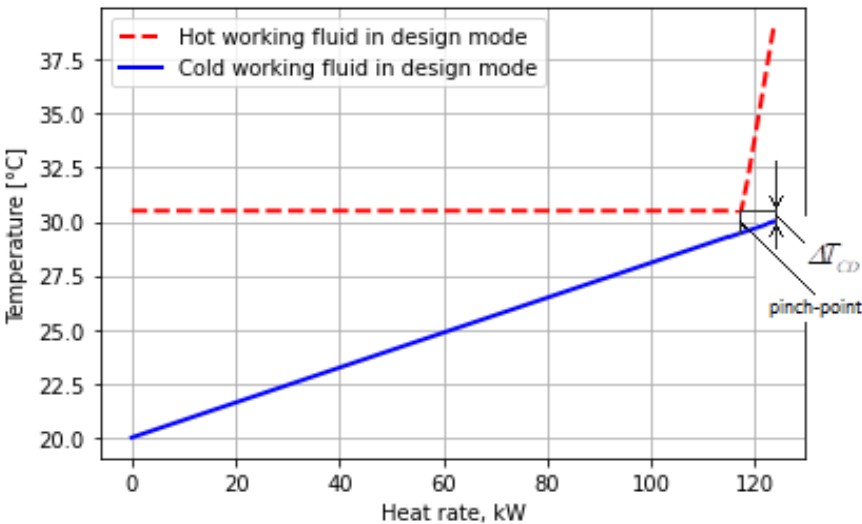

**Figure 2.** Q-T diagram of the condenser for the cycle with unavoidable parts of the exergy destruction.

Generally, it could be concluded that the results of the advanced exergy analysis for the vapor-compression refrigeration machine using R134a obtained by the authors and presented in [2] are quite similar.

The results obtained from the existing advanced exergy analysis with the ones evaluated on the basis of the new proposed approach are compared in Table 9. It could be concluded that $\dot{E}_{D,k}^{AV,EN}$, $\dot{E}_{D,k}^{AV,EX}$, $\dot{E}_{D,k}^{AV,INT}$, and $\dot{E}_{D,k}^{AV,EXT}$ of the evaporator and throttling valve coincided. However, large differences for the compressor and condenser were observed. According to the proposed approach, the value of internally caused exergy destruction within the compressor ($\dot{E}_{D,CM}^{AV,INT}$) due to irreversibilities taking place in this component was equal to 6.507 kW, which is larger by 70.4% compared to the avoidable

endogenous exergy destruction ($\dot{E}_{D,CM}^{AV,EN}$) obtained on the basis of the existing methodology. Moreover, the compressor's avoidable externally caused exergy destruction ($\dot{E}_{D,CM}^{AV,EXT}$) was equal to 3.058 kW, being 8.9% lower compared to the avoidable exogenous exergy destruction ($\dot{E}_{D,CM}^{AV,EX}$). According to the proposed approach, the condenser featured 4.575 kW of avoidable exergy destruction due to irreversibilities taking place within this component. The existing methodology suggested that 3.790 kW of the exergy destruction within the condenser can be avoided by improvement of this component. Thus, the relative difference of 20.7% was observed for the condenser. The proposed approach estimated 1.180 kW of avoidable exergy destruction within the condenser, which could be removed by improving the efficiencies of the remaining components. The existing methodology of advanced exergy analysis evaluates this part of exergy destruction equal to 1.527 kW.

**Table 9.** Comparison of the results of the avoidable of exergy destruction rates obtained by authors and presented in [2].

| Component | Author's Data $\dot{E}_{D,k}^{AV,INT}$, kW | Morosuk and Tsatsaronis [2] $\dot{E}_{D,k}^{AV,EN}$ ,kW | $\delta$, % | Author's Data $\dot{E}_{D,k}^{AV,EXT}$ , kW | Morosuk and Tsatsaronis [2] $\dot{E}_{D,k}^{AV,EX}$ ,kW | $\delta$, % |
|---|---|---|---|---|---|---|
| CM | 6.507 | 3.819 | 70.400 | 3.058 | 3.355 | −8.900 |
| CD | 4.575 | 3.790 | 20.700 | 1.180 | 1.527 | −22.700 |
| TV | 0.000 | 0.000 | - | 5.769 | 5.857 | −1.500 |
| EV | 4.356 | 4.356 | 0.000 | 0.000 | 0.000 | - |

The avoidable internally caused exergy destruction $\dot{E}_{D,CM}^{AV,INT}$ or the part that can be reduced by improving the efficiency of the compressor equal to 6.507 kW (see Table 9) can be obtained in a real cycle of the investigated refrigeration machine after increasing the isentropic efficiency of the compressor to 0.95 and performing a recalculation with the help of the service presented in [24]. As a result, exergy destruction within the compressor is decreased from 8.272 kW to 1.765 kW or by 6.507 kW. The application of the existing methodology would have suggested a lower decrease of 3.819 kW in exergy destruction for this case (see Table 9). Moreover, after improving the isentropic efficiency of the compressor from 0.80 to 0.95 and reducing the pinch point temperature difference in the evaporator from 10 K to 0.5 K on the web-page presented in [24], a decrement from 7.444 kW to 6.264 kW in the exergy destruction within the condenser could be observed. In this case, the decrease in exergy destruction was equal to 1.180 kW, which completely corresponds to the findings acquired on the basis of the proposed approach (see Table 9). According to the existing methodology of advanced exergy analysis, this improvement would enable a 1.527 kW decrease in exergy destruction within the condenser.

As explained above, within the existing methodology of advanced exergy analysis, avoidable exogenous and avoidable endogenous exergy destruction were calculated with Equations (5) and (6). It could be concluded that these parts of exergy destruction cannot be obtained without estimating the endogenous $\dot{E}_{D,CM}^{EN}$ and unavoidable endogenous $\dot{E}_{D,CM}^{UN,EN}$ exergy destructions first. However, within the existing approach, it was impossible to estimate the "pure" endogenous parts of exergy destruction within the components due to incomplete thermodynamic "idealization" within the evaporator and the condenser. Only the pinch point temperature differences equal to zero could be accepted. For such conditions, exergy destruction rates within these heat exchangers were not equal to zero—irreversibilities were not completely removed [2,14]. As a result, the endogenous and unavoidable endogenous exergy destructions for the compressor, which were equal to 4.850 kW and 1.031 kW, respectively, were not "pure" (i.e., they included some exergy destruction within the evaporator and the condenser). This drawback was afterwards implicitly included in the values of avoidable endogenous and exogenous parts' exergy destruction ($\dot{E}_{D,CM}^{AV,EN}$ and $\dot{E}_{D,CM}^{AV,EX}$). This could explain the estimation of such a large difference (70%) between appropriate values of the avoidable parts of exergy destruction for the

compressor obtained on the basis of the existing methodology of advanced exergy analysis and within the proposed approach.

When estimating $\dot{E}_{D,CD}^{EN}$ and $\dot{E}_{D,CD}^{UN,EN}$, some amount of irreversibility within the evaporator was included. As a result, this inclusion affected the values of avoidable endogenous and exogenous parts' exergy destruction ($\dot{E}_{D,CD}^{AV,EN}$ and $\dot{E}_{D,CD}^{AV,EX}$) for the condenser.

A similar explanation could be applied for the evaporator.

## 4. Conclusions

The paper presents a new approach for calculating the avoidable parts of exergy destruction within the components of energy conversion systems. The new method does not need to consider the ideal state of a component, which is one of the major drawbacks of the previous approaches.

An open-source, web-based, interactive tool for the implementation of both the conventional advanced exergy analysis and the one involving the novel approach to a refrigeration system has been developed.

It has been shown that the model developed within the work is acceptable for the comparison of the results for the exergy analysis of the investigated system. Moreover, it has been found that the outcomes obtained from the existing advanced exergy analysis and the proposed approach are quite different for the evaluated case study. The major inequality is associated with the compressor and the condenser. The existing methodology provides values of 3.819 kW for the compressor and 3.790 kW for the condenser regarding avoidable parts of exergy destruction, which could be reduced by improving the efficiency of the considered components, while the proposed approach estimates this part of exergy destruction to be equal to 6.507 kW and 4.575 kW for the compressor and the condenser, respectively. Furthermore, in the case of the application of the existing advanced exergy analysis, the avoidable parts of exergy destruction for the compressor and the condenser being removable by removing the irreversibilities within the remaining components are equal to 3.355 kW and 1.527 kW, respectively, which is higher as compared with the findings obtained by the proposed approach (3.058 kW and 1.180 kW, respectively).

In future research, the proposed approach, supplemented with similar open-source, web-based, interactive tools, will be developed for other energy conversion systems.

**Author Contributions:** Conceptualization, V.V., P.G. and E.N.; methodology, V.V., P.G. and E.N.; software, V.V. and N.B.; formal analysis, V.V. and P.G.; investigation, V.V., P.G. and E.N.; resources, V.V., N.B. and E.N.; data curation, V.V.; writing—original draft preparation, V.V.; writing—review and editing, P.G.; visualization, N.B.; supervision, V.V.; project administration, V.V. All authors have read and agreed to the published version of the manuscript.

**Funding:** This research received no external funding.

**Institutional Review Board Statement:** Not applicable.

**Informed Consent Statement:** Not applicable.

**Data Availability Statement:** The implemented open web-based interactive tool can be found at: https://colab.research.google.com/drive/1ID95CKOMqimLruhvKtSxGm4roL5Np3LH. (accessed on 21 September 2021).

**Conflicts of Interest:** The authors declare no conflict of interest.

### Nomenclature

| | |
|---|---|
| $e$ | Exergy per unit mass, $kJ \cdot kg^{-1}$ |
| $\dot{E}$ | Exergy rate, kW |
| CD | Condenser |
| CM | Compressor |
| EV | Evaporator |
| in | Inlet |
| $\dot{m}$ | Mass flow rate, $kg \cdot s^{-1}$ |
| out | Outlet |
| $p$ | Pressure, bar |
| $T$ | Temperature, K or °C |
| TV | Throttling valve |
| $x$ | Parameter obtained from the model developed by the authors |
| $x_{[2]}$ | Parameter presented in [2] |
| *Subscripts and superscripts* | |
| AV | Avoidable |
| AV, EN | Avoidable endogenous |
| AV, EX | Avoidable exogenous |
| D | Destruction |
| F | Fuel |
| EXT | External |
| INT | Internal |
| k | k-th component |
| MIN | Minimum |
| P | Product |
| PH | Physical |
| UN | Unavoidable |
| UN, EN | Unavoidable endogenous |
| UN, EX | Unavoidable exogenous |
| *Greek symbols* | |
| $\delta$ | Relative difference, % |
| $\Delta$ | Difference |

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
