# Peer review of "Simulation and Exergy Analysis of a Refrigeration System Using an Open-Source Web-Based Interactive Tool—Comparison of the Conventional Approach and a Novel One for Avoidable Exergy Destruction Estimation"

_applsci, doi:10.3390/app112311535_

Round 1

Reviewer 1 Report

Line 72-73 can be reconstructed as: unavoidable endogenous and exogenous, and avoidable endogenous and exogenous parts.

Line 74- check out the repetitive words 

Line 76- check for repetitive words

Line 91 - remove double spacing and introduce a comma

Line 94 - remove "is"

Author Response

Dear Reviewer,

thank you for the opportunity to revise our manuscript “Simulation and exergy analysis of a refrigeration system using an open-source web-based interactive tool – comparison of the conventional approach and a novel one for avoidable exergy destruction estimation”. We appreciate the careful review process and constructive suggestions. It is our belief that the manuscript has been improved after adopting the suggested enhancements. We have revised the manuscript accordingly with your suggestions and provided specific answers below.

Our responses can be found below in red text, including how and where the text has been modified. Changes made in the manuscript are marked using track changes. The revision has been developed in consultation with all co-authors, and each author has given approval to the final form of this revision.

Sincerely,

authors of the manuscript

Response to Reviewer 1 Comments

Point 1: Line 72-73 can be reconstructed as: unavoidable endogenous and exogenous, and avoidable endogenous and exogenous parts.

Point 2: Line 74- check out the repetitive words.

Point 3: Line 76- check for repetitive words.

Response 1: The authors thank the Reviewer for this suggestion. According to the advanced exergy analysis, the exergy destruction of each component belonging to the investigated energy conversion system can be split into the following eight parts: avoidable, unavoidable, endogenous, exogenous, unavoidable endogenous, unavoidable exogenous, avoidable endogenous and avoidable exogenous. Each of this part has its own meaning and should be differentiated. We tried to change the text for clearer understanding.

Point 4: Line 91 - remove double spacing and introduce a comma.

Response 2: The authors thank the Reviewer for this correction. Double spacing has been removed and a comma has been introduced.

Point 5: Line 94 - remove "is".

Response 3: The authors thank the Reviewer for this correction. “is” has been removed

Reviewer 2 Report

This paper proposed an approach to estimate the avoidable exergy destruction rates of a refrigeration system, it fits the scope of this issue and the findings are interesting. I would like to recommend acceptance if the authors can address my comments below.

  1. As there are many abbreviations across the manuscript, a Nomenclature will be useful;
  2. Page 5. Line 218-220, it’s a bit confusing that the min temperature difference is more than 10K, as there seem at least 25 degrees (-5 degree ~ 20 degree);
  3. IN Equation 8, should be 100% instead of 100;
  4. Page 6 Line 251, should be equation 9 instead of 8;
  5. What does PH mean in equation 8, please clarify it, also calrify what is the value for T0 and P0;
  6. Please use the same format for all tables, a three-line format is preferable;
  7. In Table 2, some difference estimations were incorrect, please re-check them and correct them;
  8. In table 9, some difference percentages are quite big, for example exceeds 70%, please add more explanation and clarify if they are acceptable.

Author Response

Dear Reviewer,

thank you for the opportunity to revise our manuscript “Simulation and exergy analysis of a refrigeration system using an open-source web-based interactive tool – comparison of the conventional approach and a novel one for avoidable exergy destruction estimation”. We appreciate the careful review process and constructive suggestions. It is our belief that the manuscript is improved after adopting the suggested enhancements. We have revised the manuscript accordingly with your suggestions and provided specific answers below.

Our responses can be found below in red text, including how and where the text has been modified. Changes made in the manuscript are marked using track changes. The revision has been developed in consultation with all co-authors, and each author has given approval to the final form of this revision.

Sincerely,

authors of the manuscript

Responses to Reviewer 2 Comments

Point 1: As there are many abbreviations across the manuscript, a Nomenclature will be useful.

Response 1: The authors thank the Reviewer for this suggestion. The Nomenclature is included in the end of the manuscript

Point 2: Page 5. Line 218-220, it’s a bit confusing that the min temperature difference is more than 10K, as there seem at least 25 degrees (-5 degree ~ 20 degree)

Response 2: The authors thank the Reviewer for this suggestion. The following assumptions were made [2]: (i) in the evaporator air was cooled from -5 °C down to -15 °C; (ii) in the condenser ambient air was heated from 20 °C up to 30 °C; (iii) the minimal temperature differences within the evaporator (i.e. between air and refrigerant) and within the condenser (i.e. between refrigerant and air) was 10 K.

Point 3: IN Equation 8, should be 100% instead of 100

Response 3: The authors thank the Reviewer for this suggestion. The equation (8) has been modified.

Point 4: Page 6 Line 251, should be equation 9 instead of 8

Response 4: The authors thank the Reviewer for this suggestion. The number has been changed.

Point 5: What does PH mean in equation 8, please clarify it, also clarify what is the value for T0 and P0;

Response 5: The authors thank the Reviewer for this suggestion. PH means the physical exergy. It is clarified in the text and in the Nomenclature. The T0 and P0 are dead state temperature and pressure, taken as the ambient (outdoor) ones and equal to 20 °C and 0.1 MPa, respectively.

Point 6: Please use the same format for all tables, a three-line format is preferable

Response 6: The authors thank the Reviewer for this suggestion. The same three-line format for all tables has been applied.

Point 7: In Table 2, some difference estimations were incorrect, please re-check them and correct them

Response 7: The authors thank the Reviewer for this suggestion. Corrections have been made.

Point 8: In table 9, some difference percentages are quite big, for example exceeds 70%, please add more explanation and clarify if they are acceptable.

Response 8: The authors thank the Reviewer for this suggestion. Within the existing methodology of advanced exergy analysis avoidable exogenous and avoidable endogenous exergy destruction were calculated with the following formulas

                                              ED,kAV,EN= ED,kEN- ED,kUN,EN            (5)

                                               ED,kAV,EX= ED,kAV- ED,kAV,EN            (6)

It could be concluded that these parts of exergy destruction cannot be obtained without estimating the endogenous ED,kEN and unavoidable endogenous ED,kUN,EN exergy destructions first.

However, within this existing approach it was impossible to estimate the “pure” endogenous parts of exergy destruction within the components due to incomplete thermodynamic “idealization” within the evaporator and the condenser. Only the pinch point temperature differences equal to zero could be accepted. However, for such conditions exergy destruction rates within these heat exchangers were not equal to zero – irreversibilities were not completely removed. As a result, the endogenous and unavoidable endogenous exergy destructions for the compressor, which were equal to 4.850 kW and 1.031 kW, respectively, were not “pure” (i.e., they included some exergy destruction within the evaporator and the condenser). This drawback was afterwards implicitly included in values of avoidable endogenous and exogenous parts exergy destruction (ED,CMAV,EN and ED,CMAV,EX). This could explain the estimation of such a big difference (70 %) between appropriate values of avoidable parts of exergy destruction for the compressor obtained on the base of the existing methodology of advanced exergy analysis and within the proposed approach.

As estimating ED,CDEN and ED,CDUN,EN  some amount of irreversibilities within the evaporator was included. As a result, this inclusion affected the values of avoidable endogenous and exogenous parts exergy destruction (ED,CDAV,EN and ED,CDAV,EX) for the condenser.

The similar explanation could be applied for the evaporator.

The values of avoidable parts of exergy destruction were calculated directly by thermodynamic improvement of the appropriate component. The results completely correspond to the findings made on the base of the proposed approach. This fact confirms the correctness of the developed approach.